# *InstanceAnimator*: Multi-Instance Sketch Video Colorization

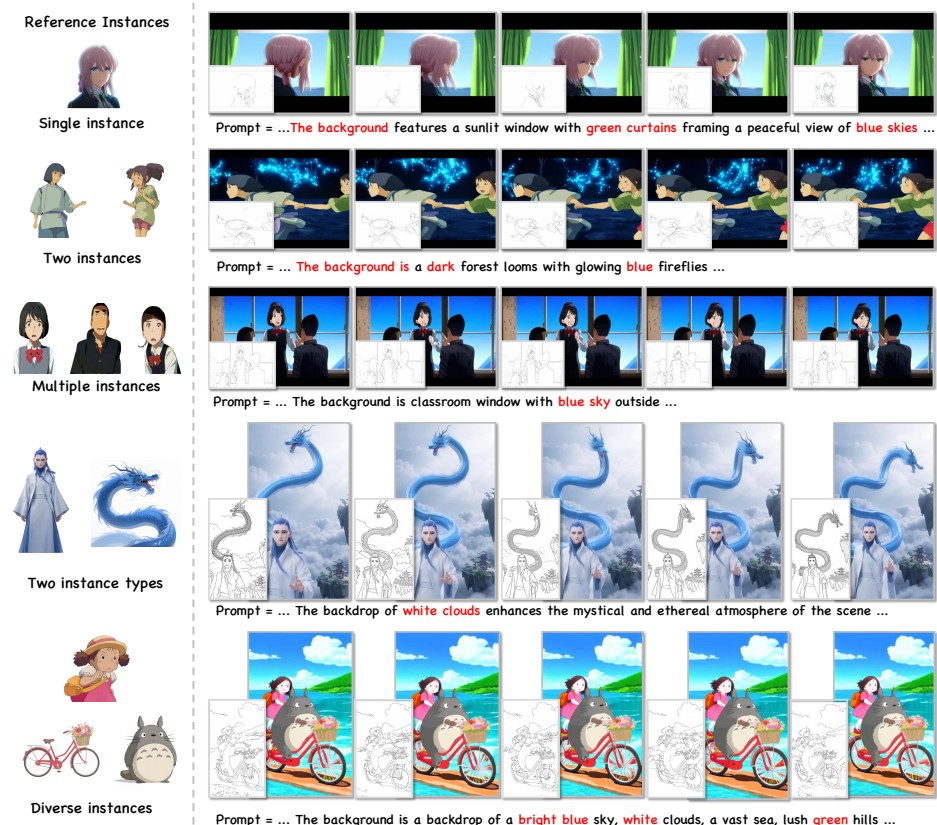

Figure 1: **Example results of InstanceAnimator**. Given diverse instances, sketch sequences, and textual descriptions, our framework enables high-quality, controllable video colorization with multi-instance and background customization.

## ABSTRACT

We propose *InstanceAnimator*, a novel Diffusion Transformer (DiT)-based framework for multi-instance sketch video colorization. Existing animation colorization methods rely heavily on a single initial reference frame, resulting in fragmented workflows and limited customizability. To eliminate these constraints, we introduce a Canvas Guidance Condition that allows users to freely place reference elements on a blank canvas, enabling flexible user control. To address the misalignment and quality degradation issues of DiT-based approaches, we design an Instance Matching Mechanism that integrates the instances with the sketch and noise channels, ensuring visual consistency across different sequences while maintaining controllability. Additionally, to mitigate the degradation of fine-grained details, we propose an Adaptive Decoupled Control Module that injects semantic features from characters, backgrounds, and text conditions into the diffusion model, significantly enhancing detail fidelity. Extensive experimental results demonstrate that *InstanceAnimator* effectively enables better user control in multi-instance colorization, producing high-fidelity results with strong temporal consistency.

## 1 INTRODUCTION

Animation plays a crucial role in visual content creation and is widely applied in education, cinema, entertainment, and digital media creation to convey coherent, dynamic narratives. As shown in Figure 2, the colorization of animation is commonly built on a meticulous and iterative process, including character design, keyframe production, frame inbetweening, and per-element colorization. Although traditional multi-stage colorization methods have allowed fine-grained control by these stages (Isola et al., 2017; Li et al., 2021; Shen et al., 2022; Cen et al., 2025; Chen et al., 2025), they still demand intensive artistic labor, especially for multi-character scenes. This multi-stage approach also requires the sketching and colorization of hundreds of frames for only a few seconds of animation, imposing significant costs in time and human effort.

Recent advances in the diffusion models (Rombach et al., 2022; Peebles & Xie, 2023) have significantly enhanced the quality and flexibility of content generation. Based on these pretrained models, researchers proposed a new approach to automate animation colorization using reference images to guide color propagation (Tang et al., 2025; Li et al., 2025; Meng et al., 2024; Yang et al., 2025). However, the core issue with these methods is their heavy dependency on a single reference frame as an exemplar, which prevents the dynamic customization capabilities that professional animators require. This rigid constraint also hinders artists from performing creative tasks, such as independently modifying a character's costume color, adjusting background lighting to fit plots and scenarios, or experimenting with alternative texture schemes. In previous methods, such changes require reprocessing the entire scene, which is inconsistent with the basic practices in production and common requirements from animators—**the iterative refinement of discrete visual elements**. Artists are thus commonly forced to sacrifice either expressiveness or efficiency in their creative process. We summarize the limitations of existing video line art colorization methods as follows:

(1) ***First frame dependency***: Current methods' reliance on a single initial reference frame causes three major limitations. First, it restricts creative flexibility by preventing independent modification of visual elements (e.g., character costumes, lighting). Second, it fails to handle large character motions (*e.g.*, running, jumping) or camera movements (e.g., panning shots), where the fixed reference becomes inadequate for drastically different poses or viewpoints. Third, it limits results to short sequences, reducing practical applicability in real production settings.

(2) ***Mismatch between instances and sketches***: Existing methods, particularly Diffusion Transformer (DiT)-based frameworks, cannot establish reliable correspondence between reference instances (*e.g.*, character design sheets) and sketch sequences. Unlike UNet-based models that leverage pixel-level feature matching to enforce spatial alignment, DiTs lack mechanisms for fine-grained visual correspondence, often resulting in unstable or inconsistent appearance.

(3) ***Detail degradation***: Existing methods often fail to preserve fine-grained visual details during propagation. Elements such as background textures, small props, clothing patterns, and subtle facial expressions are often omitted or blurred, reducing the usability of these methods in professional workflows where such details are crucial for coherence, fidelity, and expressiveness.

To address these limitations, we propose *InstanceAnimator*, a novel method for multi-instance sketch video colorization. As shown in the Figure 2, our approach is designed to address the rigidity of prior exemplar-driven pipelines, enabling flexible scene construction, robust instance consistency, and fine-grained detail preservation. Unlike traditional pipelines that rely on a fixed first frame as reference, we propose a **Canvas Guidance Condition** that allows users to freely place pre-designed reference elements (*e.g.*, characters, background components) onto a blank canvas. This canvas acts as a spatio-temporal anchor, enabling more natural propagation of appearance across sequences, particularly under large-scale character motions or camera movements. To tackle the mismatch between the reference instances and sketch sequences, we introduce an **Instance Matching Mechanism**. Specifically, we concatenate instances with the sketch, compelling the attention module to learn their visual relationships and form a stronger conditional prior. Furthermore, we inject instances into the noise channels to enhance controllability during inference, allowing text guidance to better modulate the generated content. Together, these designs ensure consistent appearance transfer while preserving the inherent controllability of diffusion models. To mitigate the degradation of fine-grained details, we propose an **Adaptive Decoupled Control Module** that selectively injects the semantic features of characters (*e.g.*, clothing patterns, subtle facial expressions) and backgrounds (*e.g.*, props, textures) into the denoising process. This module strengthens fidelity at a granular level, enabling the model to preserve intricate details that are often lost in previous pipelines.

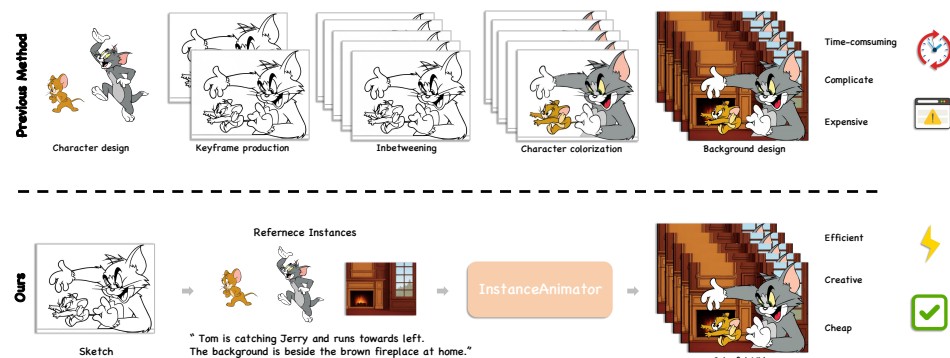

Figure 2: **Motivation**. Unlike traditional methods that require multi-stage, frame-by-frame colorization, *InstanceAnimator* supports colorizing sketch sequences into videos directly after background and character design, providing higher flexibility and significantly reducing time consumption.

As shown in Figure 1, with these designs, our framework can effectively generate high-quality results with diverse instance controls. Overall, our contributions can be summarized as follows:

- We propose *InstanceAnimator*, the first multi-instance sketch video colorization framework, to address complex multi-reference animation tasks that are ubiquitous in creative scenarios.
- We introduce **Canvas Guidance** to enhance user creativity with improved colorization quality and multi-instance with background control, and **Instance Matching Mechanism** to strengthen instance consistency by establishing robust correspondence between references and sketches.
- We present an **Adaptive Decoupled Control Module** that enables fine-grained control over text, background, and instances. We also conduct comprehensive experiments to evaluate our approach, demonstrating higher-quality, more robust results than previous methods.
- To advance the field of full decoupled control for animation workflow, we release the *OpenAnimate Dataset* as open-source, which comprises over 40K high-quality video clips along with their associated instances, background, reference frame, and detailed textual descriptions.

## 2 RELATED WORK

### 2.1 VIDEO DIFFUSION MODEL

In recent years, video diffusion models have evolved rapidly to tackle the complexity of spatiotemporal content generation, with architectural choices playing a crucial role in their performance (Ho et al., 2020; Blattmann et al., 2023a). Early advances focused on UNet-based architectures (Blattmann et al., 2023b; Ye et al., 2023; Zhang & Agrawala, 2023; Xing et al., 2024b; Zhu et al., 2024; Hu, 2024; Ma et al., 2024; Ye et al., 2023), which extended 2D image UNets to 3D by incorporating temporal dimensions into convolutional layers. These models leveraged spatiotemporal convolutions and frame-wise attention mechanisms to capture motion dynamics, enabling the generation of short video clips with basic temporal consistency. However, they faced challenges in scaling to longer sequences due to increased computational overhead and difficulty in modeling long-range temporal dependencies. More recently, Diffusion Transformer (DiT)-based approaches have emerged as a powerful alternative (Wang et al., 2025a;c; Kong et al., 2024; Yang et al., 2024). By replacing convolutional layers with transformer blocks, DiT-based video models excel at modeling global spatio-temporal relationships through self-attention, allowing them to handle longer sequences and more complex motions. These models typically decompose videos into spatial patches across frames, treating temporal dependencies as part of the attention context, which enhances their ability to preserve coherence across dynamic scenes. Despite their strengths, both UNet and DiT-based video diffusion models primarily focus on general video generation (Cao et al., 2025; Hu, 2024; He et al., 2024; Wang et al., 2024) and lack specialized mechanisms for domain-specific tasks like sketch video colorization. Such tasks require strict adherence to sketch structures, instance-level color consistency, and preservation of fine details, which cannot be achieved by general video

generation models. This research gap highlights the need for specialized adaptations tailored to the unique constraints of video line art processing.

## 2.2 REFERENCE-BASED LINE ART COLORIZATION

Reference-based colorization methods (Zhuang et al., 2024; Huang et al., 2024; Xing et al., 2024a; Jiang et al., 2024; Li et al., 2025; Meng et al., 2024; Zhang et al., 2025a; Chen et al., 2025) aim to transfer colors from a reference image to a target sketch, bridging the gap between artistic line work and colored rendering. Traditional approaches (Yan et al., 2025; Lee et al., 2020; Zou et al., 2019; Zang et al., 2024) relied on pixel-level matching or style transfer techniques, which struggled with complex sketches and diverse reference styles. With the development of diffusion models, recent methods (Loshchilov & Hutter, 2017; Liu et al., 2025a; Deng et al., 2025; Hu, 2024; Shen et al., 2022; Li et al., 2021; Zhang et al., 2025b; Xie et al., 2025) have adopted reference-guided generation, using cross-attention mechanisms to align sketch features with reference color information. However, these models typically depend on a single reference frame to propagate colors across sequences, which works adequately for short, static sketches but encounters significant challenges in dynamic scenarios. Specifically, when characters or backgrounds undergo substantial movements, these methods often suffer from color bleeding and inconsistent instance coloring. Moreover, while existing methods can handle single-character sketches well, they lack robust mechanisms to manage multi-instance scenarios where multiple characters or objects require distinct, consistent color schemes across frames. Current efforts for temporal coherence remain limited by fixed sequence length and difficulties in maintaining precise alignment between evolving sketch sequences and reference designs, highlighting the need for tailored multi-instance colorization frameworks.

## 3 METHOD

### 3.1 PROBLEM DEFINITION

We formalize the conditional sketch-text-image guided video colorization task as follows: given a sketch sequence $S = \{s_1, ..., s_t\}$, a text description $T$, multiple reference instances $I = \{I_1, ..., I_N\}$, and an image background $B$, our framework aims to generate colorful video frames $V_{\text{gen}} = \{v_1, ..., v_t\}$. The training objective is formulated as follows:

$$\mathcal{L} = \mathbb{E}_{t,x_0,\epsilon} \left[ \|\epsilon - \epsilon_\theta(x_t, t, S, T, I, B)\|^2 \right], \tag{1}$$

where $x_0$ denotes the ground-truth video, $\epsilon \sim \mathcal{N}(0, I)$ is the Gaussian noise, and $x_t$ is the noisy latent at timestep $t$. The denoising network $\epsilon_\theta$ is optimized to predict the noise conditioned on the sketch sequence $S$, text description $T$, reference instances $I$, and image background $B$.

To address key limitations in existing methods, we propose a novel multi-instance video colorization framework. Specifically, Sec. 3.2 establishes spatio-temporal consistency through a unified canvas that integrates instances and background, Sec. 3.3 enhances instance-sketch alignment via latent feature fusion, and Sec. 3.4 preserves fine-grained details through decoupled control injection.

### 3.2 CANVAS GUIDANCE

In existing methods, the reliance on the first frame as the sole reference severely constrains both character and background composition, preventing users from freely customizing these elements according to sketch inputs, thus limiting both scalability and creative flexibility. To address this issue, we propose the *Canvas Guidance* to enhance the spatial consistency of instances and preserve background information. As shown in Figure 4, the approach begins with initializing a blank canvas, onto which users can place multiple pre-designed instance elements (*e.g.*, characters, objects) according to their creative requirements. Then, the image background fills the next frames. This unified canvas functions as a spatio-temporal anchor, enabling the model to establish consistent temporal correlations between instances and the evolving target sequence across frames, thereby mitigating the temporal misalignment issues inherent in separate canvas encoding approaches. Let $C \in \mathbb{R}^{H \times W \times 3}$ denote the blank canvas initialized with zeros, where $H$ and $W$ match the spatial dimensions of the target sequence. Users place $N$ instance elements $\{I_1, I_2, ..., I_N\}$ onto $C$, resulting in a composite reference canvas:

$$C_{ref} = \text{Compose}(C, \{I_1, I_2, ..., I_N\}), \tag{2}$$

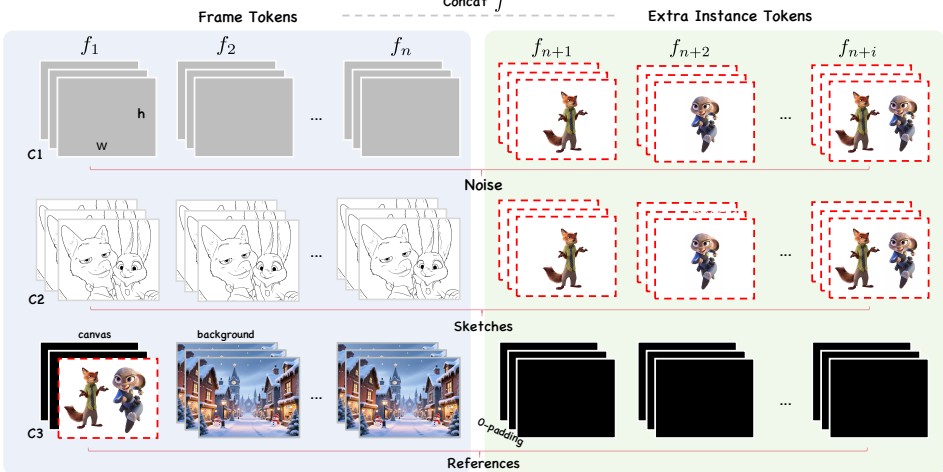

Figure 3: **Overview of *InstanceAnimator*.** We first fuse instance latent features with sketch and noise channels to establish a correspondence between the line drawing and the reference instance, as well as to maintain the character feature. Concurrently, instances, background, and text descriptions are fed into the Adaptive Decoupled Control Module independently, which dynamically injects condition information into DiT blocks through three condition-specific expert modules. At the inference stage, users can adjust the conditional weights to enhance controllability and creative flexibility.

where $\text{Compose}(\cdot)$ represents the placement operation that preserves the spatial positions and relationships of the instances. To integrate $C_{ref}$ into the diffusion process, we first encode it using a VAE encoder $E_{\text{VAE}}$ to obtain latent features in the VAE's bottleneck space: $Z_{canvas} = E_{\text{VAE}}(C_{ref}, B) \in \mathbb{R}^{H' \times W' \times D}$, where $D$ is the latent dimension and $H', W'$ correspond to the downsampled spatial dimensions of the VAE output. This latent space integration ensures that the reference canvas and target sequence frames operate in a consistent feature domain.

Figure 4: **Instance Correspondence in the DiT-based Framework.** Instance tokens are concatenated in both sketch and noise channel dimensions. This not only enables the model to capture the correspondence between reference instances and sketches, but instances can also be used as additional guiding conditions during generation without extra training parameters.

## 3.3 INSTANCE MATCHING

To address the critical need for robust correspondence between sketch sequences and reference instances and to underpin consistent visual alignment in dynamic scenarios, our *Instance Matching*

*Mechanism* enables sketch patches to attend to instance-specific features while empowering instance tokens to aggregate spatial-contextual cues. This coordinated design, paired with dynamic adaptation capabilities, ensures reliable performance across varying multi-instance scenarios. Notably, instance tokens do not participate in the calculation of losses. As shown in Figure 4, we first encode reference instances to capture their distinct visual attributes. For $N$ reference instances $\{I_1, I_2, ..., I_N\}$, we use the VAE encoder $E_{\mathrm{VAE}}$ to generate instance-specific latent features, which act as tokens for aggregating spatial-contextual cues:

$$Z^i_{\mathrm{inst}} = E_{\mathrm{VAE}}(I_i) \in \mathbb{R}^{H' \times W' \times D}, \quad \forall i \in \{1, 2, ..., N\}, \tag{3}$$

where $D$ is the latent dimension, and $H'$, $W'$ align with the spatial dimensions of sketch and canvas features. Next, we fuse these components into a unified joint feature to enable cross-modal interaction. For a given timestep $t$, let $Z^t_{\mathrm{sketch}} \in \mathbb{R}^{H' \times W' \times D}$ denote the sketch latent, and $Z_{\mathrm{canvas}} \in \mathbb{R}^{H' \times W' \times D}$ denote the canvas latent (providing global semantic priors from Canvas Guidance). As shown in Figure 4, the joint feature is defined as:

$$\underbrace{Z^{(t)}_{\mathrm{joint}}}_{\mathbb{R}^{D_j}} = \left[ \underbrace{Z^{(t)}_{\mathrm{noise}}}_{\mathbb{R}^{D_n}} \| \underbrace{Z^{(t)}_{\mathrm{sketch}}}_{\mathbb{R}^{D_s}} \| \underbrace{Z_{\mathrm{canvas}}}_{\mathbb{R}^{D_c}} \right], \tag{4}$$

where $Z^t_{\mathrm{noise}} \in \mathbb{R}^{H' \times W' \times D}$ is the noise latent, $D_j$ is the total dimension of the joint feature, and $\|$ denotes channel-wise concatenation. This $Z^t_{\mathrm{joint}}$ serves as input to DiT and the self-attention module, where $Q$, $K$, and $V$ are derived from $Z^t_{\mathrm{joint}}$ itself. This design explicitly facilitates sketch patches attending to instance-specific features, while instance tokens aggregate spatial-contextual cues from sketch regions and global priors from the canvas:

$$\mathrm{Self\text{-}Attention}(Z^t_{\mathrm{joint}}) = \mathrm{softmax}\left( \frac{QK^T}{\sqrt{D_k}} \right) V, \tag{5}$$

where $D_k$ is the qkv dimension. This process strengthens correspondence by encoding semantic and spatial relationships between sketches, instances, and global context. As shown in Figure 5(L), a core advantage of this design is its instance adaptation. Our framework inherently supports arbitrary $N$ (adding, swapping, or replacing instances) without retraining. By treating instance tokens as modular conditioning signals, the self-attention module automatically adjusts to new instances, ensuring robust generalization across diverse instances. More details are shown in Appendix B.

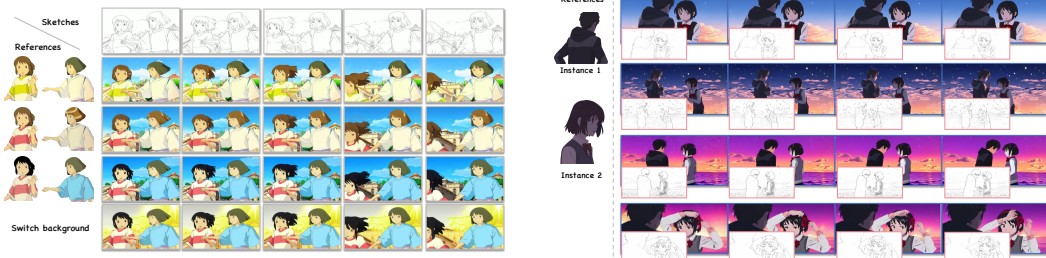

Figure 5: **Instance Control Ability.** **(L)** Given the same sketch and different reference instances, *InstanceAnimator* generates a variety of colorful videos. **(R)** Using the same designed characters, our framework colorizes different sketches with consistent colors and user-customized backgrounds.

### 3.4 ADAPTIVE DECOUPLED CONTROL MODULE

To address the critical issue of fine-grained detail degradation in background elements, instance-specific features (*e.g.*, small props, clothing patterns), and text descriptions, we propose the *Adaptive Decoupled Control Module*. This module explicitly injects detailed semantic cues into the diffusion process through four coordinated components: information projection modules, expert modules, a learnable dynamic condition weight, and a condition gating layer, ensuring the adaptive preservation of intricate details across diverse scenarios. The core design leverages CLIP and T5 encoders for

robust feature extraction, followed by modality-specific projection and targeted injection via separate cross-attention branches. Its implementation is detailed as follows: First, we encode reference background, instance elements, and text descriptions using modality-specific encoders to capture their distinct semantic and visual details. Let $bg$ denote the reference background, $\{I_1, I_2, ..., I_N\}$ denote $N$ instance elements, and $T$ denote the text description. Their raw features are encoded as:

$$F_{\text{bg}} = \text{CLIP}_{\text{vis}}(bg), F_{\text{inst}}^i = \text{CLIP}_{\text{vis}}(I_i) \ (\forall i \in \{1, 2, \ldots, N\}), F_{\text{text}} = \text{T5}_{\text{enc}}(T). \quad (6)$$

Next, to align these heterogeneous features with the hidden dimension of the Diffusion Transformer backbone, the information projection module processes each modality through a dedicated multi-layer perceptron (MLP) branch to preserve modality-specific details:

$$F_{\text{bg}} = \text{MLP}_{\text{bg}}(F_{\text{bg}}) \in \mathbb{R}^{d \times D_{\text{DiT}}},$$
$$F_{\text{inst}}^i = \text{MLP}_{\text{inst}}(F_{\text{inst}}^i) \in \mathbb{R}^{d \times D_{\text{DiT}}} \quad (\forall i \in \{1, 2, ..., N\}), \quad (7)$$
$$F_{\text{text}} = \text{Text\_Projection}(F_{\text{text}}) \in \mathbb{R}^{d_{T5} \times D_{\text{DiT}}},$$

where $d$ is the output dimension of CLIP's vision encoder, and $d_{\text{T5}}$ is the output dimension of T5's encoder. $D_{\text{DiT}}$ is the hidden dimension of DiT's transformer blocks, and $\text{MLP}_{\text{bg}}$, $\text{MLP}_{\text{inst}}$, Text\_Projection are lightweight, modality-specific MLPs that ensure dimension alignment while retaining fine-grained cues unique to each modality.

For adaptive decoupled control, we design separate cross-attention layers as condition experts for each projected feature, dynamically weighted by a learnable condition weight mechanism. For each DiT transformer block, the hidden states of the sketch sequence $H \in \mathbb{R}^{L \times D_{\text{DiT}}}$ (where $L$ is the sketch patch sequence length) serve as queries, with each modality's projected feature acting as keys and values in its dedicated cross-attention branch. The aggregated attention output is:

$$H_{\text{attn}} = W_{\text{bg}} \cdot \text{CrossAttn}(H, F_{\text{bg}}) + \sum_{i=1}^{N} W_{\text{inst}}^i \cdot \text{CrossAttn}(H, F_{\text{inst}}^i) + W_{\text{text}} \cdot \text{CrossAttn}(H, F_{\text{text}}), \quad (8)$$

where $W_{\text{bg}} \in \mathbb{R}^+$ (background weight), $W_{\text{inst}}^i \in \mathbb{R}^+$ (instance-specific weights for each $I_i$), and $W_{\text{text}} \in \mathbb{R}^+$ (text weight) form the learnable weight vector. During training, $W_{\text{text}}$ is frozen to 1 (preserving text-guided consistency), while $W_{\text{bg}}$ and $W_{\text{inst}}^i$ are learned end-to-end to adapt to scene-specific detail priorities.

Finally, a Condition Gating Layer adaptively fuses $H_{\text{attn}}$ with DiT's original hidden states $H$ via a sigmoid-based mechanism, ensuring context-aware detail injection:

$$H' = H + \sigma \left( \text{MLP}_{\text{gate}}(H) \right) \cdot H_{\text{attn}}, \quad (9)$$

where $\sigma(\cdot)$ is the sigmoid function and $\text{MLP}_{\text{gate}}$ is a lightweight MLP. As shown in Figure 5(R), this module forces the DiT backbone to explicitly attend to background, instance, and text details through separate pathways in each diffusion step. By preserving condition-specific cues and adapting weights to scene complexity, the module retains fine-grained details regardless of instance count or scene diversity, significantly enhancing the fidelity of colorized results. Example results of semantic background control are presented in Appendix Sec. C.

## 4 EXPERIMENT

### 4.1 IMPLEMENTATION DETAIL

**Experiment settings**. Our approach is based on the Wan2.1-1.3B and Wan2.1-14B (Wang et al., 2025a) pre-trained weights and trained on Sakuga42M dataset (Pan, 2024) and animation films data. The data construction pipeline is detailed in Appendix A. Training utilizes the AdamW optimizer with a learning rate of 2e-5, employing a batch size of 1 on an A800 80GB GPU. **Evaluation metrics**. We evaluate performance along multiple dimensions. FID (Unterthiner et al., 2018) quantifies video quality and natural motion smoothness. CLIP Score (Radford et al., 2021) measures semantic correspondence between the generated video and the reference video. For conditional generation tasks involving reference videos, we report Learned Perceptual Image Patch Similarity (LPIPS) (Zhang et al., 2018) and Structural Similarity Index (SSIM) (Wang et al., 2004) to assess frame-level fidelity. Temporal Consistency (Temporal) evaluates video consistency in the temporal dimension by calculating the average similarity between adjacent frames.

Table 1: Quantitative comparisons with baselines. **Bolded**: best, underscored: second best.

| Model | FID ↓ | SSIM ↑ | LPIPS ↓ | Temporal ↑ | CLIP ↑ |
|---|---|---|---|---|---|
| ToonCrafter(+First and Last Frame) | 134.497 | 0.661 | 0.272 | 0.961 | 0.894 |
| Anidoc(+First Frame) | 305.382 | 0.343 | 0.637 | 0.961 | 0.691 |
| LVCD(+First Frame) | 138.828 | 0.561 | 0.451 | 0.962 | 0.871 |
| LayerAnimate(+First Frame) | 184.048 | 0.601 | 0.431 | 0.959 | 0.849 |
| Two-Stage Animateion(+First Frame) | 245.172 | 0.424 | 0.466 | 0.954 | 0.756 |
| ToonComposer(+First Frame) | 132.319 | 0.581 | 0.304 | 0.968 | **0.929** |
| **Ours(+Instances)** | **121.901** | **0.681** | **0.201** | **0.969** | 0.921 |

## 4.2 QUANTITATIVE COMPARISON

To systematically evaluate the video colorization performance of our approach, we constructed a rigorous test set consisting of 100 high-quality animation video clips with multiple reference instances. For each clip, we manually extracted character instances as reference design images through SAM, ensuring consistent evaluation conditions across all methods. We compare our framework against four recent colorization methods: LVCD (Huang et al., 2024), ToonCrafter (Xing et al., 2024a), Anidoc (Meng et al., 2024), LayerAnimate Yang et al. (2025), and ToonComposer (Li et al., 2025). To ensure fair comparison while respecting each method's original design, we followed their default input configurations: For LVCD, Anidoc, LayerAnimate, and ToonComposer, we provided the first frame of the original video as the reference image. Two-Stage Animation includes using MangaNinja Liu et al. (2025b) to colorize the first frame and then using Wan-I2V-14B to colorize the sketches. In contrast, our method uniquely utilizes extracted reference instances for colorizing sketch sequences. As shown in Table 1, ToonComposer achieves the best in CLIP Score, while our approach achieves the best performance best in the other metric. Notably, despite not leveraging the full information of the reference frame, our method provides decoupled instances and background controllability and creative flexibility for users in sketch video colorization.

## 4.3 QUALITATIVE COMPARISON

As depicted in Figure 6, our method demonstrates superiority in generating high-fidelity results. It not only produces significantly clearer textures, such as the fine details of clothing patterns and hair strands, but also better preserves the character's unique identity, including subtle facial features and postures. In scenarios where there are substantial differences between the reference instances and input sketches, our method shines. For example, when the input sketch has a large-scale motion transformation compared to the reference, LVCD fails to accurately colorize the sketches, and it cannot follow the large-scale motion in the results, leading to disjointed color mappings. ToonCrafter, on the other hand, struggles when provided with multiple reference instances. It often misinterprets the correct color information, resulting in color conflicts or inappropriate color assignments.

Compared to Anidoc, our method demonstrates advantages in handling complex motion scenarios. Anidoc shows weakness when the reference instances themselves contain motion and when trying to follow camera trajectories. In such cases, Anidoc's results often lack coherence, with characters or background elements appearing misaligned or discolored. Moreover, a practical limitation of the competing methods is that they only support short video generation with fixed frame rates and resolutions. In today's diverse application scenarios, which demand flexibility in video length, frame rate adaptation, and resolution scalability, our method overcomes these restrictions to enable more widespread and practical usage.

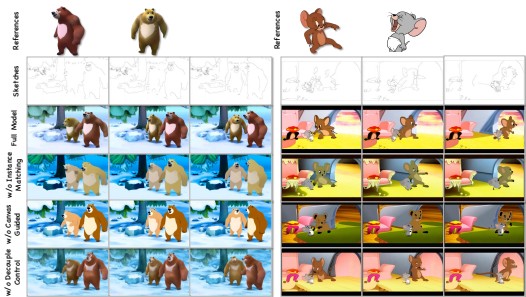

Figure 7: **Qualitative results of ablation study**. All proposed modules effectively contribute to improving the quality and consistency of multi-instance colorization.

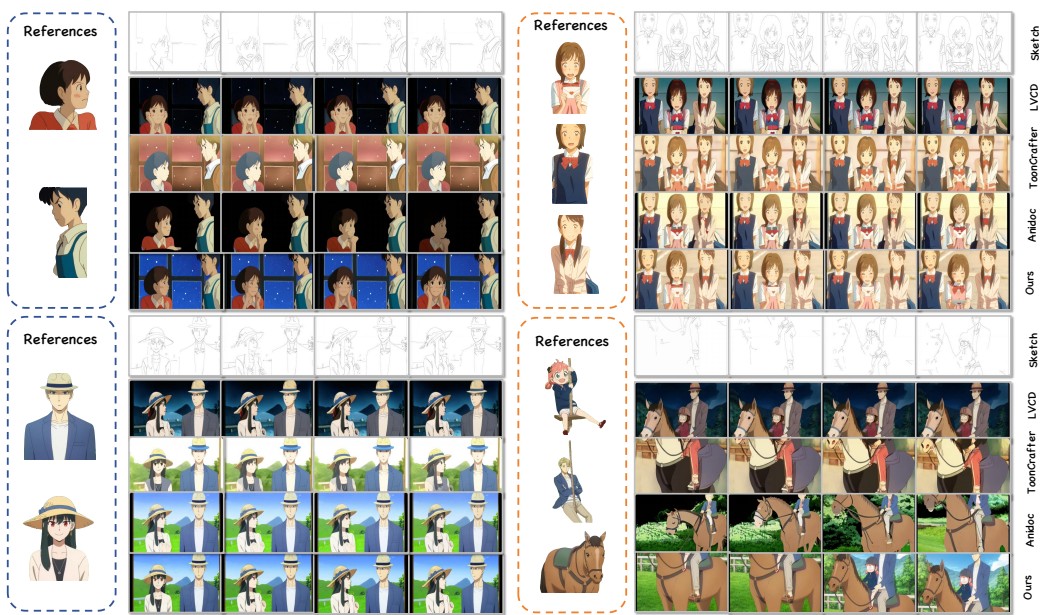

Figure 6: **Qualitative comparison with baseline methods**. Given diverse reference instances, our method achieves better results in maintaining character controllability and color consistency.

## 4.4 ABLATION STUDY

To investigate the contribution of key components in our framework, we conduct systematic ablation experiments by sequentially removing three core modules: Instance Matching, Canvas Guidance, and Adaptive Decoupled Control. As shown in Table 2, the Full Model achieves the best overall performance across most metrics, confirming the effectiveness of our integrated design.

**Effectiveness of Instance Matching:** As shown in Figure 7, the degradation is particularly evident in identity preservation results, revealing that characters lose consistent color mapping across frames, with frequent mix-ups between reference instances. This confirms that Instance Matching is critical for maintaining per-character identity throughout the video.

Table 2: Quantitative results of ablation study.

| Configuration | FID ↓ | SSIM ↑ | LPIPS ↓ | Temp ↑ | CLIP ↑ |
|---|---|---|---|---|---|
| Full Model | **121.901** | **0.681** | **0.201** | **0.969** | **0.921** |
| w/o Instance Matching | 138.615 | 0.576 | 0.317 | 0.952 | 0.861 |
| w/o Canvas Guidance | 129.221 | 0.605 | 0.289 | 0.958 | 0.875 |
| w/o Decoupled Control | 126.984 | 0.591 | 0.293 | 0.961 | 0.862 |

**Effectiveness of Canvas Guidance:** Removing this module results in higher FVD and lower SSIM compared to the Full Model. Visually, this leads to noticeable color inconsistency, especially in background regions and instances of style. **Effectiveness of Adaptive Decoupled Control:** This ablation causes more modest performance drops but introduces distinct detail degradation issues. As shown in Figure 7, fine-grained details such as facial expressions, fabric textures, and accessory patterns become blurred or inconsistent. We also include detailed functional comparisons with other methods in Appendix Sec. D and a user study in Appendix Sec. E.

## 5 CONCLUSION

In this paper, we propose *InstanceAnimator*, a novel DiT-based framework for multiple reference instances sketch video colorization. Our method includes Canvas Guidance Condition and Instance Matching Mechanism to establish a correspondence between line art sequences and reference instances and unify reference instances on a blank canvas as a spatio-temporal anchor to resolve temporal misalignment issues. Meanwhile, the proposed Adaptive Decoupled Control Module injects fine-grained background and instance details into the diffusion process, preserving intricate features like textures and patterns. We demonstrate through extensive experiments that our approach supports diverse instances and improves model generalization in the colorization process.

## ETHICS STATEMENT AND REPRODUCIBILITY

All procedures performed in studies involving human participants were in accordance with the ethical standards of the institutional and/or national research committee. This article does not contain any studies with human participants performed by any of the authors. Informed consent was obtained from all individual participants included in the study. We will release code under **CC-BY-NC-4.0**.

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

APPENDIX

## A  DATASET AND PIPELINE

**Automated data pipeline construction.**  The data pipeline is designed to automatically process raw data into structured training samples, with three core modules, as follows:

**Video Processing:** Long-form videos (e.g., full-length films, short dramas) first undergo scene segmentation to generate video clips with consistent scene contexts. For each segmented video clip, the first 81 frames are allocated as training data, and the remaining frames are reserved as extra reference frames.

**Image Processing:** A single frame is randomly sampled from each video clip (encompassing both training frames and extra reference frames) to serve as the reference frame. The reference frame is processed in three steps:

- Step 1: Instance Extraction. Each foreground instance is extracted using the Segment Anything Model (SAM) Kirillov et al. (2023), Recognized Anything Model Zhang et al. (2023), and Grounding Dino Liu et al. (2024).
- Step 2: Character Generation. Qwen-Image-Edit Wu et al. (2025) is employed to generate complete character images based on the extracted instances.
- Step 3: Background Generation – Additionally, complete background images are obtained via text-guided editing with Qwen-Image-Edit by removing foreground characters while retaining background details.

**Text Processing:** Finally, the textual description for each video clip is constructed to cover four mandatory dimensions. *Location:* spatial position of each character in the video. *Appearance:* detailed appearance attributes of each character. *Motion:* actions performed by each character throughout the video. *Background:* contextual description of the video background, ensuring comprehensive semantic guidance for model training.

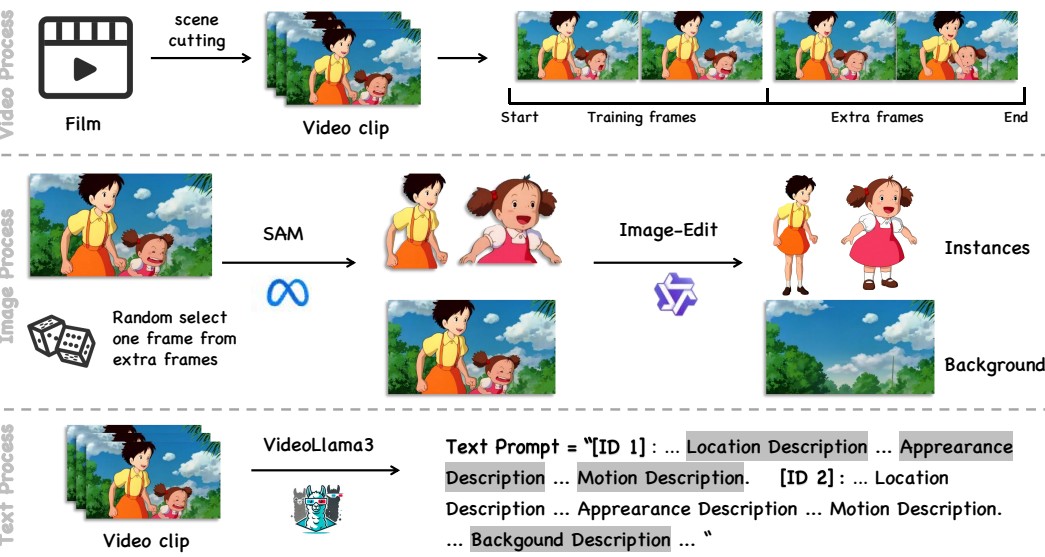

Figure 8: **Data Construction Pipeline**. We build the complete training data in three steps: Video, Image, and Text Processing.

**OpenAnimate Datset**  This dataset is designed for the instance-aware sketch video colorization task and aimed at promoting the development of automatic animation techniques. This dataset includes two parts. One is a filter subset from the Sakuga42M dataset, the other is collected from animation films on the internet. The total video clips are about 40K+, and each video clip is in the

same scene, which is cut by scene-cutting algorithms. For each data point, we provide a reference frame, multiple references, a background image, and a text description.

## B  INSTANCE CONTROL

In the animation video colorization task, instance control can be categorized into two distinct scenarios:(i) The reference instance and the target sketch character belong to different identities, leading to substantial visual discrepancies that may hinder accurate alignment;(ii) The reference instance and the sketch character share the same identity but differ in pose or attire, where reliable colorization results are generally achievable.

Recent works have explored identity preservation for text-to-video generation tasks. ConsistID Yuan et al. (2025) utilizes Q-Former to fuse facial features and keypoint features for enhancing identity representation. At the same time, CineMaster Wang et al. (2025b) integrates class labels, 3D bounding boxes, and comprehensive textual descriptions, leveraging ControlNet to strengthen identity control. However, these methods suffer from inherent limitations: Q-Former requires large-scale training data due to its fixed-dimensional learnable queries and exhibits limited expressive capacity, whereas ControlNet replicates entire DiT blocks, incurring substantial training parameter overhead.

Distinct from these approaches that leverage Q-Former or ControlNet for text-visual feature fusion, we propose a direct MLP-based Adaptive Decouple Control Module, delivering a more concise and computationally efficient cross-modal alignment paradigm. Our method's instance control capability is anchored in two complementary core mechanisms: Visual Reference: Entailing the learning of correlations between reference instances and input sketches to establish a direct visual correspondence; Textual Guidance: Seamlessly integrated into our data pipeline design. Specifically, during caption construction, we explicitly describe each instance's spatial location and detailed appearance attributes in natural language, providing precise semantic cues for alignment. Within the Adaptive Decouple Control Module, we conduct cross-modal fusion of text and image features. Leveraging these fused features, which encapsulate both visual structural information and textual semantic constraints, we mitigate character misalignment to the maximum degree, ensuring accurate instance-specific colorization.

As shown in Fig. 9 and 10, we present diverse reference instances across distinct styles to validate our model's instance control capability. Specifically, in Fig. 9 (single-instance control scenario), we vary the reference instances from the same identity to completely distinct identities. Our generated results consistently adapt to changes in the visual references in a noticeable manner, reflecting precise instance alignment. For Fig. 10 (multi-instance control scenario), we extend the setting to multiple reference instances whose identities and styles are fully inconsistent with those of the input sketches. Even under this challenging condition, our results still demonstrate strong generalization by transferring the style of the references to the colorization output. Notably, when identities and styles are completely mismatched, our model primarily leverages the overall style of the references rather than strict pixel-wise color alignment. This is because one-to-one visual correspondence between references and sketches is absent, making precise color matching infeasible.

## C  BACKGROUND CONTROL

For background control, there are two significant things to consider: background consistency and controllability. As shown in Figure 11, our method enables custom background control during the colorization process, which operates both at the semantic and visual level. Unlike naive approaches that need other tools to directly replace the background, our control mechanism prioritizes alignment with the sketch's inherent structure. It both provides text-driven semantic control without providing a background image and directly visual background image control, ensuring the colorized background not only matches the sketch's visual logic but also avoids unnatural discrepancies between the background and foreground. Semantic background control is advantageous in controllability; it can generate static or dynamic background, and it is expedient to modify according to user intention. Visual background control is better in consistency; it fully follows the user-provided background.

However, when multiple reference characters are combined with complex character movements, the effect of background control is very likely to have flaws or color overflow. On the one hand, it

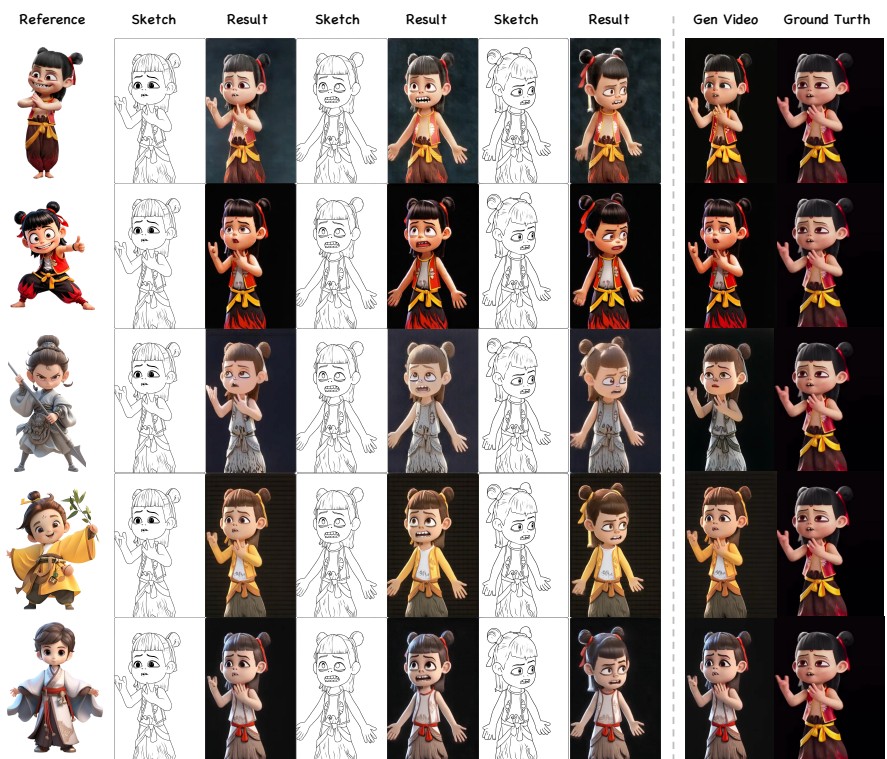

Figure 9: **Colorization from Different Style Reference**. For the same character, our method easily learn features. For different style characters, our method can transfer their style to the sketches even without prior knowledge.

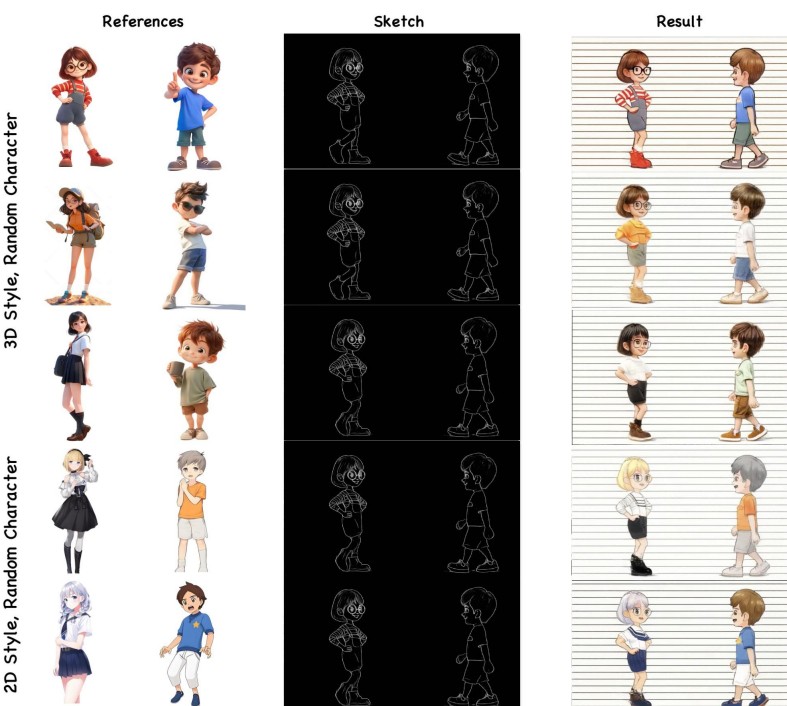

Figure 10: **Multiple Reference Instances Control**. While we obtain different style reference instances from the internet, our method can match the most similar ID and follow the reference instance style to colorize the sketch sequence.

is due to the scarcity of high-quality line drawing data corresponding to multiple characters and backgrounds; on the other hand, it is because the model generates ambiguity when simultaneously fusing multiple character subjects and backgrounds. Besides, the generative model in DDIM steps is hard to fully preserve the original color with one hundred percent from the provided references, but it is within an acceptable range. Despite this, our method performs exceptionally well in background control for no more than four subjects and also performs well in scenes with moderate motion amplitudes. The visual background control results are provided in the supplementary materials.

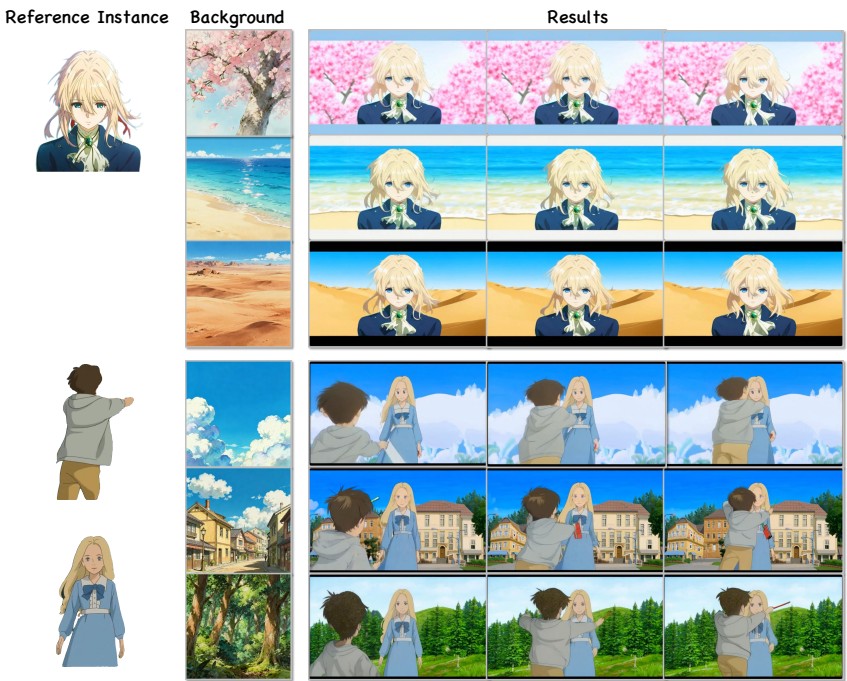

Figure 11: **Semantic background control**. Unlike traditional methods that rely on other tools for directly replacing background, *InstanceAnimator* supports semantic mapping of background in combination with line drawing during the colorization process.

## D  FUNCTIONAL COMPARISON

Table 3 presents a comprehensive comparison of our *InstanceAnimator* against existing video colorization methods across eight key functional dimensions. The comparison reveals significant limitations in current approaches: traditional methods like LVCD (Huang et al., 2024) and Anidoc (Meng et al., 2024) rely on single-frame references and struggle with multi-character scenarios, large motion handling, and advanced editing capabilities. While ToonCrafter (Xing et al., 2024a) introduces first-and-last frame guidance and ToonComposer (Li et al., 2025) achieves better multi-character support, both methods still lack canvas placement control and element-level editing functionality. In contrast, our *InstanceAnimator* demonstrates superior capabilities across all dimensions through its novel instance-based reference system. Notably, we are the only method that supports canvas placement and element editing, enabling precise control over character positioning and fine-grained modifications. The instance-based approach also ensures robust handling of large motions and maintains high temporal consistency while preserving detail quality, making *InstanceAnimator* a comprehensive solution for professional video colorization workflows.

## E  USER STUDY

To evaluate the practical performance of our method, we designed a user study comparing it with three representative baseline methods across four core user-centric dimensions. 20 participants (aged 18–60, 60% of whom are female) were recruited, all of whom had adequate knowledge and experience in computer vision, line art drawing, or related fields. This ensures participants have sufficient

Table 3: **Functional Comparison.** Detailed comparison of different video colorization approaches across key functional dimensions. ✓: full support, △: partial/limited support, ✗: not supported.

| Capability | LVCD | ToonCrafter | Anidoc | ToonComposer | Ours |
|---|---|---|---|---|---|
| Reference Type | Single frame | First+last frame | Single frame | Single frame | Instance-based |
| Multi-Character | △ | △ | △ | ✓ | ✓ |
| Canvas Placement | ✗ | ✗ | ✗ | ✗ | ✓ |
| Large Motion | ✗ | △ | ✗ | △ | ✓ |
| Element Editing | ✗ | ✗ | ✗ | ✗ | ✓ |
| Text Control | ✗ | ✓ | ✓ | ✓ | ✓ |
| Temporal Consistency | △ | ✓ | △ | ✓ | ✓ |
| Detail Quality | △ | △ | △ | ✓ | ✓ |

expertise to distinguish subtle differences in colorization quality and tool usability. All participants provided informed consent and were compensated according to local standards.

Figure 12: **User ratings across four key dimensions: Our method vs. three baselines**. All ratings are based on a 7-point Likert scale (1 = Strongly Disagree, 7 = Strongly Agree).

The study was conducted remotely by observing the results of coloring the same set of line drawings using different methods and experiencing different colorization workflows, which lasted 20 minutes per participant. We applied the generated dataset for a comparative study in the Main Sections 4.1-4.2. We randomly sampled 27 video clips from the 100 pairs of examples for each participant. Participants were presented with one video at a time (randomly shuffled) and asked to assess the quality and consistency of each video clip on 7-point scales. After this, participants experienced two interfaces of reference-based (AniDoc) and instance-based (ours) colorization workflows in random order and evaluated their usability and controllability on 7-point scales.

Based on user needs and perception goals, the four evaluation dimensions are defined as follows:

- **Quality**: Measures the visual naturalness, accuracy, and overall satisfaction of the colorized output. Participants rated their agreement with the statement: "I am satisfied with the final colorization effect."

- **Consistency**: Assesses color stability across consecutive video frames. Participants rated agreement with: "I think the colors between video frames are very consistent."

- **Usability**: Evaluates the ease of operating the workflow. Participants rated agreement with: "This workflow is easy to learn and use."

- **Controllability**: Quantifies the user's ability to adjust and refine colorization results. Participants rated agreement with: "I feel I can control the colorization results well."

We collected the results and calculated the average ratings for each method by dimension. As shown in Figure 12, while our *InstanceAnimator* ranks second only to ToonComposer in quality and consistency, our method has significantly greater efficiency, using 14 times fewer parameters and without requiring the whole first frame as a condition that ToonComposer relies upon. In terms of usability and controllability, our instance-based approach is also ahead of the reference-based approach (5.350±1.268 vs. 5.250±1.832 for usability; 5.300±1.302 vs. 5.000±1.974 for controllability) because we can achieve disentangled instance and background controls that are more in line with practical applications. Participants also conducted creative experimentation using both interfaces. Based on their feedback, the highest-rated features of our *InstanceAnimator* are the support for intuitive animation workflow, achieving longer temporal coherence with minimal reference requirements.

## F  THE USAGE OF LARGE LANGUAGE MODELS

We employed large language models (LLMs), including DeepSeek-R1[1] and Doubao[2], to support selected parts of this research. These tools were used for text refinement (including grammar checking and language polishing with DeepSeek-R1) and format optimization (including the formatting of charts and graphs with Doubao). The authors take full responsibility for the content, interpretation, and use of the LLM-generated material in this paper.

---

[1]https://www.deepseek.com/
[2]https://www.doubao.com/

