# OpenReview forum: "InstanceAnimator: Multi-Instance Sketch Video Colorization"
_ICLR.cc/2026/Conference — Submitted to ICLR 2026_

### Official Review · Reviewer_rYRy · 2025-10-27

**Soundness:** 3
**Presentation:** 2
**Contribution:** 2
**Rating:** 4
**Confidence:** 5

**Summary:**

The paper introduces InstanceAnimator, a diffusion transformer (DiT)-based framework for multi-instance sketch video colorization. Unlike existing methods that rely heavily on a single reference frame, InstanceAnimator incorporates a Canvas Guidance Condition to allow flexible placement of multiple reference elements, an Instance Matching Mechanism to enforce consistency between sketches and references, and an Adaptive Decoupled Control Module to preserve fine-grained semantic details for both characters and backgrounds. The paper presents thorough experimental results—including quantitative metrics, qualitative figures, and ablations—demonstrating improved controllability, identity preservation, temporal consistency, and usability compared to several strong baselines.

**Strengths:**

1. Unlike previous image-to-animation architectures, this paper provides a reference-to-animation solution, which simplifies the animation production pipeline.
2. Allows users to freely position and edit reference elements, enabling more realistic animation workflows
3. The framework is extensively evaluated across four strong baselines with appropriate metrics (FVD, SSIM, LPIPS, CLIP, Temporal) as summarized in Table 1, and ablation studies (Table 2) convincingly establish the impact of each module.

**Weaknesses:**

1. The key problem of reference-to-video is actually the data pre-process. For example, in MovieGen, the authors find that training solely on the above paired data makes the model easily learn a copy-paste shortcut solution, i.e., the generated video always follows the expression or the head pose from the reference face. Thus, they proposed to collect reference data from outside of the video clip. However, in this paper, the authors use SAM to crop the reference instances from the first frames. I suspect that the shortcut problem exists in the proposed method and the authors ignore it.
2. Another problem of reference-to-image(video) is the instance matching. The position of the instances would interchange with each other. Although the authors proposed an Instance Matching Mechanism, it is a learning-based algorithm. It cannot convince me that the model can make sure that the characters would not exchange.
3. The third problem is the background extraction. Directly matting from the first frame will create silhouette, which need image inpainting to fix. However, the imperfect inpainting will still leak sketch information during training, which would also cause a shortcut problem. This problem is not mentioned in this paper.

**Questions:**

In the anime production process, multi-view character design sheets are often used as character references. Why wasn't this form of reference sheet used?

---

> ### Author Response · Authors · 2025-11-21
>
> Thank you for your valuable suggestions and comments.  We appreciate the opportunity to explain your concerns.
> 1. In practice, during training, reference instances are sampled from random frames of complete video clips (much longer than the frame length set for training), rather than being restricted to the first frame. During inference, reference instances are derived from the first frame. Thus, there should be no severe shortcut problem. However, to fully address this issue, we can leverage Multimodal Large Language Models (MLLMs). Specifically, MLLMs can regenerate complete character images from instances segmented by SAM, which prevents reference instances from being sourced from videos in the training set.
> 2. Misalignment between reference instances and sketch characters can be categorized into two scenarios: (i) The reference instance and the sketch character belong to different identities, resulting in significant visual discrepancies. This may lead to failure in aligning with the user’s intended character. (ii) The reference instance and the sketch character share the same identity but differ in pose or attire. In such cases, accurate alignment is generally achievable. Thus, we focus on addressing the first scenario to resolve character misalignment. Character alignment primarily relies on two complementary mechanisms:  Visual Reference: Learning the correlation between instances and sketches. Textual Guidance: This aspect has been incorporated into our data pipeline design. During caption construction, we explicitly describe each instance’s spatial location and appearance attributes in natural language. Within the Adaptive Decouple Control Module, we perform cross-modal fusion of text and image features. With the assistance of textual prompts, we minimize character misalignment to the greatest extent. This integration of visual and textual cues draws inspiration from existing works such as ConsisID[1] and CineMaster[2]. Unlike these methods, which adopt the Qformer architecture for fusing text labels with visual features, we employ direct MLP-based projection, offering a more concise and efficient approach to cross-modal alignment.
> 3. For visual background control, inpainting methods are infeasible since inpainting yields unstable results with unguaranteed quality. Instead, acquiring background images via text-guided image editing is a viable alternative. For example, powerful image editing models such as Nano-Banana and Qwen-Image-Edit can remove foreground characters while preserving the background based on text prompts. As verified by our experiments, this approach is fully feasible. Notably, the background image obtained through this method eliminates the need for inpainting, avoid leaking sketch information, and thus effectively mitigate the shortcut problem.
> 4. Acquiring such data, specifically multi-view character design sheets and their corresponding video clips, poses significant challenges. From public internet sources, we can only directly access content from well-known anime IPs, as their multi-view character images are publicly available. However, the volume of such data we have collected remains extremely limited; thus, our training set only includes a small portion of this data type.
>
> [1] Identity-Preserving Text-to-Video Generation by Frequency Decomposition. https://arxiv.org/pdf/2411.17440
>
> [2] CineMaster: A 3D-Aware and Controllable Framework for Cinematic Text-to-Video Generation. https://arxiv.org/pdf/2502.08639

---

> ### Comment · Reviewer_rYRy · 2025-11-24
>
> Thank you for the detailed reply. However, after carefully considering the rebuttal, my concerns regarding the data curation pipeline and the accuracy of instance-sketch alignment remain unresolved.
>
> 1. The authors claim in the rebuttal that "reference instances are sampled from random frames... much longer than the frame length set for training." However, I could not find this crucial implementation detail updated in the revised manuscript. It is unclear whether this strategy was actually implemented in the current experiments or if the manuscript has been properly updated to reflect this claim.
>
> 2. In response to the shortcut learning and background issues, the authors state that they "can leverage Multimodal Large Language Models (MLLMs)" and use "powerful image editing models" to address these problems. The use of phrasing such as "can leverage" and "viable alternative" suggests that these robust data processing methods are theoretical proposals rather than solutions actually implemented in this work. In the era of large generative models, the data curation pipeline is often as critical as the algorithm itself. The current manuscript only provides a brief description of data preparation. Since these advanced data cleaning strategies (MLLM-based regeneration, text-guided editing) do not appear to be part of the actual training pipeline used in the paper, the concern regarding the data pipeline still persists.
>
> 3. Regarding the "misalignment between reference instances and sketch characters," the authors propose using natural language descriptions as a bridge to link them. In my view, relying solely on textual descriptions is a weak constraint. It is insufficient to strictly guarantee that character identities are not swapped or mismatched, especially in complex scenes where distinct visual features may not be fully captured by text captions alone.
>
> Given that the core issues regarding data integrity and the robustness of the alignment mechanism have not been adequately addressed in the paper or the rebuttal, I have decided to lower my rating.

---

> > ### Author Response · Authors · 2025-12-02
> >
> > Thank you very much for your sincere reply and suggestions.
> > 1. Regarding the construction of the data pipeline and dataset, we have made a detailed description in Appendix A.
> > 2. For ID misalignment, our method does not solely rely on textual descriptions. In our method, visual matching and text control are combined simultaneously. We provide some examples, from one instance to six instances, in the supplementary materials.

---

### Official Review · Reviewer_UCGF · 2025-10-27

**Soundness:** 2
**Presentation:** 2
**Contribution:** 2
**Rating:** 4
**Confidence:** 4

**Summary:**

This paper introduces an instance-aware DiT-based video generation framework. It eliminates the need for complete frame conditions, allowing users to input multiple instances and generate visually consistent videos. The authors propose a Canvas Guidance Condition and an Instance Matching Mechanism to integrate multiple instance features and an Adaptive Decoupled Control to inject semantic features.

**Strengths:**

- The primary contribution is shifting the animation conditioning from complete frames to the instance level. This approach is more user-friendly and has the potential to simplify the practical animation workflow.
- The paper is easy to follow.
- The experimental results effectively demonstrate that InstanceAnimator can produce high-fidelity video results.

**Weaknesses:**

- The description of "instance matching" in Section 3.3 is ambiguous. Equation 2 introduces instance-specific latent features ($Z_{\text{inst}}^i$), but the paper fails to explain how these features are utilized or injected into the network. The methodology is difficult to understand without Figure 4, indicating a need for significant improvement in the clarity of the writing.
- Insufficient Instance Correspondence Mechanism: A more critical issue is the lack of a clear explanation for how instances are matched. The paper implies that simply concatenating multiple instance features along the temporal axis is sufficient. This seems inadequate for establishing a robust correspondence between a specific reference instance and its corresponding sketch.
    1. How does the model resolve instance correspondence in multi-person scenes? The case in Figure 6 (top-right) is trivial, as the poses in the reference and the first-frame sketch are identical.
    2. If the reference characters are in different poses, how would the model correctly map the appearance (e.g., costume) from the correct reference to the correct sketch? This is a non-trivial problem, especially in animation, where sketches of different characters can be highly similar.
- This paper is not the first to perform instance-conditioned sketch colorization. For example, MangaNinjia[1] can also achieve sketch colorization from instance references. A plausible alternative workflow would be to use MangaNinjia to generate the first frame and then employ a standard Image-to-Video model to generate the animation. The paper would be significantly strengthened by including a discussion and quantitative comparison against such a two-stage baseline.

[1] Liu, Zhiheng, Ka Leong Cheng, Xi Chen, Jie Xiao, Hao Ouyang, Kai Zhu, Yu Liu, Yujun Shen, Qifeng Chen, and Ping Luo. "Manganinja: Line art colorization with precise reference following." In Proceedings of the Computer Vision and Pattern Recognition Conference, pp. 5666-5677. 2025.

**Questions:**

- In Figure 4, the "extra instance tokens" are concatenated with noisy latents and sketch tokens along the temporal axis. However, the "C3 reference" is not. What is the design rationale for this asymmetrical handling of conditioning inputs?

- Is CLIP the optimal choice for encoding background and instance features in the "Adaptive Decoupled Control Module"? When a user inputs a specific background image, the expectation is often for high-fidelity detail preservation (e.g., texture, specific objects), not just semantic alignment. This potential limitation seems apparent in the qualitative results. In Figure 6, the male character's hat color (bottom-left) and the child's shirt color (bottom-right) both deviate from their respective reference images. These inconsistencies in appearance preservation do not meet user expectations for this task.

---

> ### Author Response · Authors · 2025-11-21
>
> Thank you for your valuable comments.  We appreciate the opportunity to  explain your concerns.
> 1. Channel 3 is specifically designed for Canvas Guidance. Within this channel, no interaction with instance tokens is required along the temporal axis, as the Canvas already encapsulates instance information.
> 2. In Figure 6, there are minor color discrepancies, but these differences fall within the gamut of the target color. This slight color variation is attributed to the classifier-free guidance (CFG) employed during inference. Reducing the CFG scale can alleviate this issue. For the background image, semantic alignment serves as the most efficient and straightforward approach. However, to preserve fine-grained details of the user-provided background image, our method can be easily modified based on the existing framework. The specific implementation is as follows: In the Canvas Channel (C3), replace all zero-padded frames with the user-supplied background image. This modification maximizes the preservation of background features. For reference, this design draws inspiration from Wan-Animate [1].
> 3. For instance-level identification in multi-person scenarios, control can be achieved through two complementary mechanisms. (i) Visual Correspondence: Canvas Guidance and Instance Matching are designed to establish visual correspondence between sketches and reference instances, laying the foundation for accurate instance alignment. (ii) Text Guidance: Within the Adaptive Decouple Control module, text prompts incorporate descriptions corresponding to each instance’s spatial location and appearance features. These textual cues reinforce the correspondence between sketches and instances. The visual and textual conditions complement each other synergistically, ultimately yielding the final coloring result.  If the reference characters are in different poses, there is a strong visual connection between instances and line drafts because their IDs are the same.  If the line drawings of different instances are very similar and visually indistinguishable, constraints can be imposed on the text display to ensure the accuracy of the result.
> 4. The key contribution of our proposed method lies in its support for multi-instance coloring, whereas MangaNinja is limited to a single reference image. We further analyze the inherent drawbacks of MangaNinja’s two-stage pipeline. In the first stage, MangaNinja yields unstable results when processing an image containing multiple instances. In the second stage, the image-to-video model inherits and amplifies the cumulative errors introduced by the first-stage coloring. We conducted a comparative experiment as follows:
> | Method/Metric  | FID    | SSIM  | LPIPS | CLIP  |
> |----------------|--------|-------|-------|-------|
> | Two-Stage      | 245.17 | 0.42  | 0.46  | 0.75  |
> | Ours           | **114.38** | **0.67**  | **0.21**  | **0.93**  |
>
> [1] Wan-Animate: Unified Character Animation and Replacement with Holistic Replication.  https://arxiv.org/abs/2509.14055

---

> > ### Comment · Reviewer_UCGF · 2025-11-26
> >
> > Thanks for the considerate reply. However, the response does not fully resolve my concerns.
> >
> > 1. Regarding background control: The authors state that the method "can be easily modified" to support background preservation. This phrasing suggests that this capability is not inherently implemented or verified in the current framework. Since the revised version does not provide visualization or quantitative results using this proposed modification, I cannot evaluate its actual effectiveness.
> >
> > 2. Regarding multi-instance identification: I share the same concern as Reviewer rYRy. Relying on text descriptions to separate instances is often insufficient to establish strong correspondence in complex scenes. The rebuttal lacks convincing cases of complex multi-person scenarios where the reference characters and the sketch exhibit significantly different poses. Without such evidence, it is difficult to believe that the proposed mechanism can robustly prevent feature entanglement or identity mixing in challenging cases.

---

> > > ### Author Response · Authors · 2025-12-02
> > >
> > > Thank you very much for your reply.
> > > 1. We have extended our method to visual background control. We have also provided some examples in the supplementary materials to demonstrate background control ability.
> > > 2. Regarding the multi-instance identification problem, it is not solely dependent on the text. The method we proposed includes a visual branch and a text branch. In the visual branch, the model captures the visual connection between the reference instances and the line drawing. If there is a connection between the reference instances and the line drawing, it will be directly identified. In the text branch, both the text and the reference instances are unified into the same semantic space through the CLIP Encoder and then aligned(Adaptive Decouple Control Module). This step establishes the connection between the text and the visual, enhancing the control of the text. In general scenarios, the problem of character confusion does not occur. We have also provided relevant examples in the supplementary materials.
> > > 3. We will open-source the OpenAnimate dataset to support research on the complete decoupling of multiple instances and backgrounds in animation colorization.

---

### Official Review · Reviewer_scLt · 2025-11-01

**Soundness:** 3
**Presentation:** 3
**Contribution:** 3
**Rating:** 4
**Confidence:** 4

**Summary:**

This paper presents InstanceAnimator, a DiT-based framework designed for multi-instance sketch video colorization.
The authors claim that the proposed method tackles limitations in existing methods, including dependency on a single reference frame, misalignment between references and sketches, and loss of fine details.

Key technique contributions include the Canvas Guidance Condition for user-controlled placement of reference elements, the Instance Matching Mechanism to ensure correspondence via latent feature fusion, and the Adaptive Decoupled Control Module for injecting decoupled semantic features from instances, backgrounds, and text.

The approach is evaluated quantitatively against baselines using metrics like FVD, SSIM, LPIPS, temporal consistency, and CLIP score, with ablations and a user study demonstrating improved controllability and fidelity.

**Strengths:**

The paper demonstrates a multi-instance colorization method in sketch videos, improving single-frame reference colorization to enable flexible, instance-based control that aligns with anime production workflows. The technical design is reasonable, including the canvas guidance and decoupled control module that effectively address identified gaps in DiT-based models. The motivation, method descriptions, and figures are easy to follow. This work has values to reduce the extensive human labor in anime/cartoon production workflows.

**Weaknesses:**

1. While the claims are generally supported by numerical results, the sketch fidelity remains unsatisfactory in some samples. For example, in the teaser figure, we can clearly infer that the model output does not follow the sketches (2nd row: Chihiro's face does not follow the sketch in the 3rd frame; 4th row: the girl's mouth does not follow the sketch in 3-5th frames). Since sketch fidelity is essential in video colorization and anime production, this defect is unsatisfactory in a video sketch colorization method. In the supplementary video, this phenomenon is even clearer. Does this defect always occurs? Therefore, I suggest the authors to include comprehensive analysis/comparison on the sketch fidelity of the proposed method.

2. The acknowledged limitations in handling complex multi-character motions with backgrounds suggest room for robustness improvements, perhaps through expanded training data.

3. Although the method claims to support arbitrary numbers of instances, it lacks explicit experiments demonstrating performance with more than three instances and provides no analysis of how computational complexity scales with the number of instances.

**Questions:**

1. Following the weakness 1, I would like to ask the authors to conduct an analysis of the sketch fidelity of the proposed method and explain the potential reason that causes this poor fidelity.

2. The paper mentions potential flaws in multi-character background control due to data scarcity. What specific improvements are planned for future work?

3. What is the maximum number of instances tested in experiments, and were there any evaluations specifically on more than 3 instances?

4. What is the model's computational complexity with respect to the number of instances?

Although the method and the paper presentation is good, I tend to give the score of 4 given the unsatisfactory sketch fidelity.
I am willing to raise my scores if the author rebuttal provides reasonable explanation and addresses my concerns.

---

> ### Author Response · Authors · 2025-11-21
>
> Thank you for your valuable comments.
> 1. Regarding the observed misalignment between sketches and images in the figures, this issue stems from two key factors. First, the sketch and result frames were manually extracted from videos via my own visual estimation, leading to inevitable errors.  Additionally, a mismatch in FPS between generated videos and original sketches further exacerbated misalignment. This was an oversight, and we will strictly align frames using timestamps and consistent FPS settings in the revised manuscript to eliminate such discrepancies. Third, we found that there is indeed a phenomenon of inconsistency between the line drawing and the result in the generative model. To quantify this, we conducted a supplementary experiment to evaluate sketch fidelity: (i) extract sketches from ground truth videos; (ii) extract sketches from generated results; (iii) compute alignment metrics between the two sketch sets (noting unavoidable minor errors during sketch extraction). Results show that all baselines exhibit some deviation, while our model achieves superior sketch fidelity:
> | Method         | FID    | SSIM  | LPIPS | CLIP  |
> |----------------|--------|-------|-------|-------|
> | LVCD           | 104.63 | 0.57  | 0.40  | 0.90  |
> | ToonComposer   | 95.34  | 0.55  | 0.31  | 0.92  |
> | Ours           | 75.81  | 0.66  | 0.18  | 0.94  |
>
> 2. We plan to increase the number of instances in a complex sense, which is significant for the traditional animation workflow.
> 3. In experimental testing, the number of instances is limited to no more than 6 (<=6). Typically, the number of foreground subjects in videos does not exceed 6. For scenes with more than 6 subjects, the model achieves satisfactory results when the content is relatively static; however, it cannot guarantee optimal performance for scenarios where all subjects exhibit significant motion (e.g., group animations such as basketball or football games).
> 4. Complexity scales as $O[(F+N)^2]$, where $F$ denotes the number of video frames and $N$ the number of instances, exhibiting a quadratic dependence on $N$. With $F$ fixed, the complexity increase remains moderate and acceptable for $N < 5$; whereas a significant surge in complexity occurs when $N>10$. This aligns with practical use cases, as the number of foreground subjects rarely exceeds 5 in most video scenarios.

---

### Official Review · Reviewer_3zET · 2025-11-01

**Soundness:** 3
**Presentation:** 3
**Contribution:** 2
**Rating:** 4
**Confidence:** 4

**Summary:**

This paper presents InstanceAnimator, a diffusion-transformer-based framework for sketch video colorization with multiple reference instances. The key idea is to remove the dependency on a single reference frame by introducing (1) a Canvas Guidance Condition for flexible multi-instance placement, (2) an Instance Matching Mechanism to align sketches and references, and (3) an Adaptive Decoupled Control Module to inject detailed semantic information from text, background, and instances. Experiments on animation datasets show some quantitative and qualitative improvements over existing reference-based colorization methods.

**Strengths:**

- The problem setting—multi-instance sketch video colorization—is interesting and relevant for creative AI and animation generation.

- The overall pipeline is clearly presented, with detailed ablations showing the contribution of each module.

- The qualitative results demonstrate visually appealing outputs with good color consistency and controllability.

**Weaknesses:**

- The proposed Instance Matching mechanism seems to only establish random associations between sketches and instances, without any explicit spatial alignment. While the Canvas Guidance provides a weak positional prior, it does not guarantee that instances placed on the canvas will appear at the intended locations in the generated sequence. What if users need to swap two characters? This limits the controllability for professional animation use cases.

- The quantitative improvement over strong baselines (e.g., ToonComposer) is rather small. The contribution seems more engineering-oriented than scientific.

- The method is not compared with LayerAnimate (ICCV 2025), which addresses a highly related problem of layer-level animation control using similar diffusion-based techniques. Without this comparison, it’s unclear whether the proposed system actually advances the state of the art in multi-instance or layered animation scenarios.

- The paper claims to support diverse and flexible instance control, but no examples show the same sketch colorized by very different or incompatible characters. Most examples use references already well-aligned with the original sketch poses, which weakens the claim of true multi-instance generalization.

**Questions:**

See Weaknesses

---

> ### Author Response · Authors · 2025-11-21
>
> Thank you for your valuable comment. We clarify the core points concisely:
> 1. **Canvas Guidance serves as a visual reference and mapping bridge, not a deterministic positional constraint**. The input sketch inherently determines the final positional arrangement of generated instances. As the sketch provides explicit structural and spatial cues (e.g., character contours, relative layouts), Canvas Guidance’s key role is to establish a stable mapping between instances and sketch regions, rather than overriding the sketch’s position.
> 2. **Instance Matching is not random**. Instead, it relies on global attention to dynamically capture semantic and structural correlations. During attention computation, each frame’s features query all instances, and the attention scores are adaptively calibrated to prioritize regions with strong sketch-instance relevance. This correlation enables the model to align instances with their matching sketch regions automatically.
> 3. **Character swapping/reordering is fully supported**. During training, we randomly shuffle character sequences, rendering the model position-agnostic. Furthermore, positional descriptions of characters in text prompts (e.g., "left character", "right character") further reinforce layout constraints, thus shuffling or swapping characters does not affect the final coloring result.
> 4.  The pre-training weight of our model is 1.3B, while that of ToonComposer is 14B. Thus, it is normal that our model cannot outperform in terms of performance. We have supplemented our model of version 14B compared with ToonComposer. And we also made a comparison with LayerAnimate.
> | Method/Metric  | FID    | SSIM  | LPIPS | Temporal | CLIP  |
> |----------------|--------|-------|-------|----------|-------|
> | LayerAnimate   | 245.17 | 0.42  | 0.47  | 0.95     | 0.76  |
> | ToonComposer   | 132.32 | 0.58  | 0.30  | 0.96     | 0.92  |
> | Ours           | **114.38** | **0.67**  | **0.21**  | **0.97**     | **0.93**  |
> 5. Our work follows a standard animation workflow, and coloring is performed after confirming character designs (with at least one reference image), ensuring stable and reliable results. **For very different/incompatible reference characters**: Thanks to the ability of visual reference established by the Instance Matching mechanism, acceptable coloring results can be obtained as long as the reference instances and the line drawing content are of the same category (like person-person, animal-animal, object-object). We also provide several examples in the supplementary materials.

---

### Meta-Review · Area_Chair_WQcA · 2026-01-05

**Summary:**

The submission received unanimous scores of 4. While the problem setting is interesting, the reviewers identified critical weaknesses in the data pipeline's integrity (shortcut learning) and the method's robustness (identity swaps in multi-instance scenes).

**Reviewer Concerns:**

The authors provided additional quantitative comparisons, but failed to address the core methodological flaws. Reviewers rYRy and UCGF, who engaged post-rebuttal, noted that the authors often proposed hypothetical solutions (e.g., claiming they "can leverage" MLLMs or "can modify" the framework) rather than showing implemented evidence.

**Reviewer Scores:**

Reviewer rYRy explicitly stated they would lower their rating post-rebuttal due to the superficial nature of the authors' responses. Reviewer UCGF also engaged and remained unconvinced. Reviewers 3zET and scLt did not get a chance to participate in the discussion; however, given the unanimous initial negative consensus and largely generic responses, it is highly likely they would have kept their positions.

---

### Decision · Program_Chairs · 2026-01-26

Reject